# Pitfalls of Using NIR-Based Clinical Instruments to Test Eyes Implanted with Diffractive Intraocular Lenses

**DOI:** 10.3390/diagnostics13071259

**Published:** 2023-03-27

**Authors:** Fidel Vega, Miguel Faria-Ribeiro, Jesús Armengol, María S. Millán

**Affiliations:** 1Departament d’Òptica i Optometria, Universitat Politècnica de Catalunya, Barcelonatech, ViolinistaVellsolà 37, 08222 Terrassa, Spain; 2Physics Center of Minho and Porto Universities (CF-UM-UP), University of Minho, 4710-057 Braga, Portugal

**Keywords:** intraocular lens, diffractive lens, retinal image quality, ocular aberrations, near infrared, through focus analysis, presbyopia, cataract surgery

## Abstract

The strong wavelength dependency of diffractive elements casts reasonable doubts on the reliability of near-infrared- (NIR)-based clinical instruments, such as aberrometers and double-pass systems, for assessing, post-surgery, the visual quality of eyes implanted with diffractive multifocal intraocular lenses (DMIOLs). The results obtained for such patients when using NIR light can be misleading. Ordinary compensation for the refractive error bound to chromatic aberration is not enough because it only considers the best focus shift but does not take into account the distribution of light energy among the foci which strongly depends on the wavelength-dependent energy efficiency of the diffractive orders used in the DMIOL design. In this paper, we consider three commercial DMIOL designs with the far focus falling within the range of (−1, 0, +1)-diffractive orders. We prove theoretically the differences existing in the physical performance of the studied lenses when using either the design wavelength in the visible spectrum or a NIR wavelength (780 to 850 nm). Based on numerical simulation and on-bench experimental results, we show that such differences cannot be neglected and may affect all the foci of a DMIOL, including the far focus.

## 1. Introduction

The development of presbyopia-correcting intraocular lenses (IOLs), such as enhanced monofocal, diffractive and non-diffractive extended-depth-of-focus (EDOF) and diffractive multifocal intraocular lenses (DMIOLs), (see, for instance, [1,2,3] and the references contained therein), has paved the way for the evolution of cataract surgery to become a refractive procedure which aims to restore visual quality for distance vision and to provide patients with functional intermediate and near vision. In addition, clinicians nowadays have sophisticated instrumentation at hand, ranging from autorefractors to wavefront aberrometers and double-pass-based systems, which do not use any dazzling wavelengths in the near-infrared (NIR) spectral band (ranging between 780 and 850 nm), to objectively determine the patient’s refraction and visual quality post-surgery [4,5]. Aberrometers and double-pass systems have proven to provide a reliable clinical assessment of healthy phakic subjects’ visual function [6] by correctly accounting for the change of optical properties occurring between NIR light (used to measure) and the visible (VIS) light used in the clinics with subjective tests [7]. Reliable assessments of pseudophakic patients implanted with standard monofocal IOLs can also be obtained using NIR-based instruments [8]. However, early reports on diffractive IOLs repeatedly warned of the issues that arise from outcoming wavefront sampling (with abrupt slope changes at the diffraction zone boundaries) and the wavelength dependency of the diffraction orders that operate in the foci formation [9,10,11,12]. Thus, with Hartmann–-Shack based aberrometers there are lenslet locations in the sensor where multiple spot patterns hinder the accurate reconstruction of the DMIOL’s outcoming wavefront [10,11,12]. Moreover, DMIOLs are designed for optimum performance at a VIS wavelength of 550 nm, a wavelength at which the eye reaches the maximum photopic efficiency. Diffractive bifocals split light into the far and near foci for distance and near vision, respectively. In most diffractive bifocals, the far focus corresponds to the zeroth diffraction order (base power) whereas the near focus corresponds to the first diffraction order (with near add power). The mismatch between the design wavelength, in the VIS spectral region, and the wavelength used for testing, in the NIR region, induces a change, predicted by simple diffraction theory (see, for instance, [13]), in the energy distribution between the foci. At such, with NIR light the far focus benefitted while the near one was severely underestimated [14]. Clinicians should bear in mind that this issue may lead to further deficient or contradictory interpretation of assessments such as visual acuity defocus curves, contrast sensitivity and potential dysphotopsia (unwanted photic phenomena) experienced after surgery by the patients implanted with DMIOLS.

Some authors overlooked these principles of diffractive optics and extended the use of NIR-based instruments to determine post-surgery the visual and optical performance of patients implanted with DMIOLs (diffractive bifocals at the time) [15,16,17,18,19]. A pyramidal wavefront sensor-based aberrometer with a high spatial resolution and finer sampling has been used in recent publications [20] to overcome the aforementioned limitation of wavefront sampling of Hartmann–Shack sensor-based aberrometers. The authors evaluated the visual quality outcomes of patients implanted with nine different IOL designs (ranging from standard monofocal, enhanced monofocal, EDOF and trifocal diffractive IOLs) [21]. Strangely, in terms of distance image quality calculated from the IOLs’ measured wavefronts with a monochromatic (850 nm) Strehl ratio, the group of subjects implanted with the trifocal diffractive AT Lisa tri stood out above other groups implanted with monofocal designs [21], therefore raising concerns about whether the authors omitted the wavelength dependency of the optical performance of DMIOLs [22].

Yet in another recent paper [23], the authors report the use of a double-pass system that measures with NIR light (780 nm), to objectively assess with several metrics (MTF, PSF expressed as the Sthehl ratio) the quality of the retinal image of a group of patients implanted with a diffractive EDOF IOL (Tecnis Symfony ZXR00). These precedents indicate that some clinicians are still far from being aware of the inherent flaws in this type of assessment [24].

To make matters worse, nowadays DMIOLs that rely on non-zeroth diffraction orders for the far focus [25,26] are available on the market and thus, the use of NIR-based instrumentation with patients implanted with such *unconventional* DMIOLs may be the origin of even more misleading outcomes and may cause the clinician to reach wrong conclusions. Some authors admit a certain unreliability in testing near vision, but claim that the NIR-based assessment is still useful in distance vision [27]. However, they should take into account the fact that the wavelength dependency of the energy efficiency of DMIOLs affects all the orders of diffraction, including the zeroth order. Since the diffractive design might have a strong influence on the final outcome, the assessments obtained under NIR illumination should be cautiously excluded from the clinical evaluation of patients implanted with DMIOLs.

This paper emphasizes this warning about distance vision with a further demonstration of the physical phenomena involved in DMIOLs and illustrates the potential risk of using NIR-based equipment for assessing pseudophakic subjects implanted with DMIOLs. To this end, we consider three commercial DMIOL designs with the far focus falling within the range of (−1, 0, +1) diffractive orders. We prove theoretically the differences existing in the optical performance of such lenses when using either the design wavelength, in the VIS spectrum, or a NIR wavelength (780 to 850 nm). Based on numerical simulation and on-bench experimental results, we show that such differences cannot be ignored or neglected because they may severely affect all the foci of a DMIOL, also including far focus.

## 2. Materials and Methods

### 2.1. Intraocular Lenses

We analyzed the optical performance under VIS and NIR illumination of three market-available DMIOLs whose far focus relies, by design, on different diffraction orders, i.e., m = 0 (AT Lisa Tri), m = +1 (Tecnis Symfony) and m= −1 (sinusoidal Acriva Trinova). Table 1 summarizes their most relevant features for our study.

The AT Lisa Tri 839MP IOL (Carl Zeiss Meditec AG, Jena, Germany) with +3.33D and +1.66D add powers for the near and the intermediate foci, respectively, is made of hydrophilic acrylic (25%) material with hydrophobic surfaces. The lens has a 1.46 refractive index (at the design wavelength, 550 nm) and a 56.5 Abbe number [28]. The aspheric design of the IOL is intended to produce a negative value of spherical aberration (SA) of −0.18 μm (6.0 mm entrance pupil) to compensate for the natural positive SA of the human cornea [29]. Multifocality is achieved by means of a diffractive anterior surface of 6 mm diameter. At the center, the trifocal zone has a diameter of 4.34 mm. The outer region of the lens, to the 6 mm edge, is bifocal and splits light into the far and near foci exclusively. More in detail, the trifocal zone features the combination of two bifocal diffractive profiles [30]. The first one uses the m = 0 and m = +1 diffraction orders to allocate energy in the far and intermediate foci (+1.66D add power at 550 nm), respectively, while the second profile uses the m’ = 0 and m’ = +1 diffraction orders to split the energy into the far and near foci (+3.33D add power at 550 nm) [31]. Thus, the AT Lisa Tri IOL is used in this work as an example representative of the DMIOLs that operate in the far focus with the 0th diffraction order. So far, most of the DMIOLs (bifocal and trifocal) launched on the market operate with the 0th diffraction order in far vision as well.

The Tecnis Symfony IOL (model ZXR00, Johnson & Johnson Vision, Groningen, The Netherlands) is made of hydrophobic acrylic material with a refractive index of 1.47 (at the design wavelength of 550 nm) and an Abbe number of 55 [32]. As with other diffractive IOLs in the Tecnis family, the anterior lens surface is aspheric while the diffractive profile is engraved in the posterior spherical surface of the lens. Overall, the Symfony IOL is designed to correct +0.27 μm of corneal SA (6.0 mm entrance pupil) [33]. The diffractive pattern is formed by nine rings (referred to as echelletes by the manufacturer) [34] to split the light into two foci (far and intermediate), separated by relatively low addition (+1.75 D) as shown in different optical bench experiments [35,36]. A thorough study by Millan and Vega [25] demonstrated that the Tecnis Symfony is a hybrid refractive–diffractive IOL that operates at the higher harmonic diffraction orders m = +1 and m = +2 to form the far and intermediate foci of the lens, respectively. Once implanted, this design allows for chromatic aberration compensation in both foci, thus contributing to enhancing the imaging properties of the eye’s optical system.

The Acriva Trinova lens (VSY Biotechnology, Amsterdam, The Netherlands) is a hybrid refractive–diffractive trifocal IOL made of hydrophilic and hydrophobic acrylic copolymer material. The refractive index (at the design wavelength of 550 nm) is 1.46 and the Abbe number is 58 [37]. The features of its sinusoidal diffractive profile have been reported in an earlier work [26]. Thus, the sinusoidal diffractive part is located on its anterior surface and consists of a central disk—also referred to as the first diffractive zone—plus 11 concentric rings. The posterior surface is purely refractive and, according to the manufacturer, the asphericity of the lens provides a ‘mild correction’ of SA (−0.165 μm) [38]. Unlike the trifocal diffractive designs that rely on the combination of two bifocal diffractive profiles (e.g., the AT Lisa Tri’s), a single sinusoidal diffractive profile, such as the Acriva Trinova’s, operates with three diffraction orders at the same time, m = −1, 0 and +1, for the far, intermediate and near foci, respectively [39,40]. The add powers of the Acriva Trinova at 550 nm are +1.5 D for intermediate focus and +3.0 D for near focus.

In this study, we considered DMIOLs of the three different designs but with the same power (20D) for the far focus, meaning that far vision was attempted with either the m = −1 (Acriva Trinova) or m = 0 (AT Lisa Tri) or m = +1 (Tecnis Symfony) diffraction order.

### 2.2. Simulated Optical Performance with VIS and NIR Metrics

The theoretical assessment of the optical imaging quality of the three IOLs was simulated using custom-written software, implemented in Matlab (Mathworks, Natick, MA, USA). The software makes use of Fourier Optics physical principles to calculate the distribution of light at different planes along its direction of propagation after the light has been refracted/diffracted by the surfaces and apertures of a model eye with specifications set to replicate the ISO 2 model eye, with this being experimentally implemented in an optical bench as described in the next section.

We used the data from the diffractive phase profiles obtained by confocal microscopy, as reported by several authors: the Symfony’s and Trinova’s profiles, both measured with a PLμ confocal microscope (Sensofar, Terrassa, Spain), from Millán et al. [25] and Vega et al. [26], respectively; AT Lisa Tri’s profile, measured with a MarSurf CWM 100 confocal microscope (Mahr GmbH, Göttingen, Germany), from Frey et al. [30]. Other parameters were calculated to replicate each IOL optical design as accurately as possible, namely their chromatic properties and corneal SA correction, specified in Table 1. These parameters were used to reconstruct synthetic models of the IOLs. The software starts by defining the optical path difference for a collimated beam, at the wavelength used to illuminate, referenced at the exit pupil of the optical bench setup (see full description in the next section). A high density of points (1 µm lateral resolution) is used to assure accurate sampling of the diffractive profile, including realistic transition zones between adjacent diffractive steps that will contribute to the background intensity captured by the experimental setup. This can be achieved with high computational efficiency by taking advantage of the circular symmetry of the system. The through-focus procedure is conducted by adding defocus wavefronts to the distance-focused wavefront, with values ranging from +3.0 D to −6.0 D, in 0.05 D steps. This procedure is equivalent to scanning the image plane, as experimentally implemented and described in the next section. At each step, the far-field point spread function (PSF) is calculated as the absolute square value of the quasi-discrete Hankel transform [41] of the generalized circularly symmetric pupil function [42].

Point spread functions (PSF) were calculated for each IOL at all defocus positions. The calculations were repeated for 3 mm and 4.5 mm physical apertures at the IOL plane, and for 530 nm, 780 nm and 850 nm wavelengths. Next, the through-focus (TF) PSFs were convolved with the image of a 200 µm- pinhole object. We computed the energy efficiency (EE) in each focus, which is the experimental approximation of the theoretical light-in-the-bucket (LIB) metric [13,43]. The EE metric evaluates the diffraction energy efficiency at the foci as well as the image blur caused by out-of-focus images, aberrations and scattering. We calculated the energy within the central core of the pinhole image (E_core_, gray level summation of the pixels of the zone) and the total image energy (E_total_, gray level summation extended to the pixels of the entire image). The energy of the full image comprises the core and the background (E_total_ = E_core_ + E_background_). The EE is the amount of light energy in the core relative to the energy in the full image, that is the ratio (E_core_/E_total_). The simulated EE value was plotted versus defocus position, aperture diameter and wavelength.

### 2.3. Experimental Setup for Optical Performance Assessment with Visible and NIR Light

The optical imaging quality of the IOLs was assessed using an optical test bench with a model eye (artificial cornea with diaphragm and wet cell) which has been described in detail in former works [44,45,46] and is summarized here. Figure 1 shows a sketch of the setup which consists of three parts: the illumination system, the model eye and the image acquisition system.

Two narrow-band light sources were used: a VIS (green) and a NIR light emitting diodes (LEDs) (Thorlabs GmbH, Munich, Germany) with emission centered at 530 nm and 780 nm, respectively. They both had a full-width half-maximum spectral bandwidth of 32 nm, which represents 6% (at 530 nm) and 4% (at 780 nm) of the central emission of the LEDs. Either the VIS or the NIR LED illuminated the test object located at the front focal plane of a collimator (200 mm focal length). The tests were a 200 µm pinhole for the assessment of the energy distribution among the foci planes (far and near), and a four-slit pattern for modulation transfer function (MTF) measurements [47].

The collimated beam illuminated the model eye with the IOL immersed. An iris diaphragm was used as the entrance pupil (EP) to control the size of the beam on the artificial cornea and hence, the level of corneal SA of the wavefront that impinged upon the IOL under test. Additionally, the EP size also determined the beam size at the IOL plane (referred hereafter to as IOL-pupil). For instance, for a collimated wavefront and a 6.0 mm EP, the cornea of the Liou–Brennan model eye would send a converging beam onto the IOL that exposes a diameter of 5.15 mm at the IOL plane [48].

The model eye type 2, as specified in ISO 11979-2:2014 [49], was implemented in our optical bench. The artificial cornea was an achromatic doublet (Lambda-X, Nivelles, Belgium) that produced a physiological amount of 4th-order SA of +0.16 μm (5.15 mm IOL-pupil). Therefore, taking into account the SA values of the studied IOLs (Table 1), it could be argued that the Acriva Trinova IOL would benefit (unfairly) when its optical performance is evaluated since its negative SA perfectly compensates for that of the artificial cornea, thus leaving the model eye virtually free of SA. However, we highlight that even with the largest pupil (6.0 mm entrance pupil, 5.15 mm IOL pupil), the maximum remnant SA of the model eye would be only −0.02 microns and −0.09 microns for AT Lisa Tri and Tecnis Symfony, respectively. Moreover, since the maximum pupil diameter considered in this work was 4.5 mm, even smaller remnant SA values [50] were expected, and therefore, differences in the optical performance of the IOLs related to differences in the SA compensation can be discarded. For further details about the clinical relevance of the model eye, refer to [33,34,35,36,37,38,39,40,41,42,43,44,45,48].

The optical performance of the IOLs with both green and NIR illumination was evaluated through-focus at the image space of the model eye. The aerial images formed by the model eye with the IOL under test were projected through a 10× infinity corrected microscope (Olympus Plan Achromat, Wells Research, Inc., West Covina, CA, USA) onto a monochrome 8-bit CCD camera (Wells Research, Inc., West Covina, CA, USA). We checked that, for both wavelengths, the CCD sensor response was linear in the dynamic range of interest. To improve the signal to noise ratio, each image was eventually the result of averaging eight image frames. The microscope and camera set were moved along the bench axis to locate the best focal planes for each IOL with a spatial resolution of 1 µm. Axial scanning was stretched to smoothly cover the focal segment of interest from distance to intermediate and near images.

We recall that according to the ISO 11979-2 standard, with the model eye type 2 the vergence of light entering the eye cannot be varied and must be kept in collimation. The rationale for this guideline is that the power of model eye type 2 is lower than the natural power of the human eye and thus, their magnifications are not comparable for objects at finite distances. Then, instead of varying the object vergence, the camera image plane must be changed by a distance that is set by reference to the best focus for far vision. However, it is very important to note, given the conjugation relationship between the object and image spaces that applies for any optical system, that it is possible, with geometrical optics, to derive the relationship between the distances that are required to move the camera image plane for a specified entering object vergence (see, for instance, Equation C.10 in American National Standard for Ophthalmics—Extended Depth of Focus Intraocular Lenses, ANSI Z80.35-2018). Then, with our model eye illuminated with a collimated beam, the physical TF scan carried out in the image space was equivalent to cover an object vergence range of about 9 D (from +3.0 D to −6.0 D) in 0.1 D steps with 0.05 D resolution. From now on, the images recorded with wavelengths of 530 nm and 780 nm will be referred to as green and NIR images, respectively.

For each IOL, the origin for defocus (0.0 D) was experimentally set according to the ISO 11979-2:2014 recommendation, i.e., at the image plane where, with green light and 3.0 mm IOL-pupil, the maximum MTF at 50 cycles/mm was obtained. This plane turned out to be the best focus plane of the IOL for distance vision. Figure 2 shows the pinhole image formed by the Acriva Trinova under green light at the distance focus plane, or equivalently, 0.0 D defocus.

In brief, the image in a given focal plane consists of the core sharp image of the pinhole surrounded by a blurred and faint halo-shaped background (Figure 2a). The halo is clearly visible when a proper gamma correction is applied to the image (Figure 2b). The practical implementation of the EE metric versus the theoretical LIB requires the ideal point source to be replaced by a pinhole of a certain size, which was described and justified in a former work [51]. To compute the EE, an edge detection algorithm (Canny edge detector implemented in MATLAB MathWorks) was firstly applied to segment the central core from the image of the pinhole. Since the gray level of a pixel was proportional to the energy impinging on that pixel, we computed the energy of the light reaching the core (E_core_) by integrating the gray level of the pixels within the region. The energy of the image as a whole (E_total_) was likewise calculated from the pixels of the entire image. The EE metric was then estimated from the experimental ratio E_core_/E_total_ [14,25]. The uncertainty in the computed values of the energy was basically due to the precision in the determination of the size of the core. Assuming an uncertainty of ±1 pixel in the diameter of this region of interest, the highest relative error corresponded to the lens focus with the lowest EE and smallest IOL-pupil (3.0 mm), which proved to be lower than 8% for the three IOLs.

The through-focus energy efficiency (TF-EE) of the IOLs was measured sequentially with green and NIR illuminations. It is of note that there was a local maximum of EE in all the IOL’s foci (distance, intermediate and near).

## 3. Results

### 3.1. Through-Focus Energy Efficiency (TF-EE)

Figure 3 shows, for the three IOLs (Acriva Trinova, AT Lisa Tri and Tecnis Symfony) and the 4.5 mm IOL-pupil, the experimental (Figure 3a–c) and simulated (Figure 3d–f) results of the TF-EE metric obtained with green and NIR illumination. The results with the 3.0 mm IOL-pupil were similar and are provided in the Appendix A. The excellent correspondence between experimental and simulated TF-EE curves, for both the green and the NIR wavelengths, should be noted. This high agreement is further confirmed by the quantitative results contained in Table 2, which summarizes, for each particular IOL and wavelength, the EE of the foci and the power difference between the most distant EE peaks (i.e., far versus near peaks in the case of Acriva Trinova and AT Lisa Tri, and far versus intermediate peaks for Tecnis Symfony).

The drastic effects of the illumination change—from VIS (green) to NIR—can be seen immediately from these results. Thus, while Acriva Trinova IOL (Figure 3a,d) shows a balanced energy distribution among its three foci under green illumination, with EE values around 0.30 (Table 2), it shows a prominent intermediate focus (EE around 0.5) with NIR at the expense of the far (0.2) and near (0.2) foci.

In the case of AT Lisa Tri IOL (Figure 3b,e), the far focus of the lens benefits considerably from NIR illumination, while the near and, more significantly, the intermediate foci show a severe reduction of their EE. Despite the changes, the trifocal nature of both DMIOLs, the Acriva Trinova and AT Lisa Tri, can be still recognized from either TF-EE curve (green or NIR), although with markedly different add powers.

With regard to Tecnis Symfony IOL (Figure 3c,f), the change from green to NIR illumination causes a variation in the lens performance, in terms of EE, that is even more severe. On the one hand, with green light, the IOL behaves as a bifocal low-addition lens, with two uneven EE peaks, corresponding to the far and intermediate foci, 1.7D apart (1.8D according to the simulated results). On the other hand, with NIR illumination, the experimental and simulated TF-EE curves show a single and larger EE peak for far vision, thus accounting for a monofocal behavior. No additional tiny peak was detected at −2.5D defocus, where an intermediate focus with 780 nm NIR light was predicted (Table 2), rather just a subtle decrease in the slope was apparent. Therefore, it can be said that, under NIR illumination, Tecnis Symfony IOL becomes a monofocal lens.

Finally, it is worth noting that, when changing from green to NIR illumination, the power difference between the extreme EE peaks or foci increases for all three IOL designs.

### 3.2. Halo Assessment

To acquire further insight into the differences in the optical performance of the IOLs between green and NIR illumination, Figure 4, Figure 5 and Figure 6 show the experimental pinhole images at the corresponding foci of the lenses, i.e., at the defocus position labelled as Far (*F*), Intermediate (*Int*) and Near (*N*) in Figure 3. The corresponding simulated images are shown in the Appendix A.

These experimental and simulated images of the pinhole test give even more graphic evidence of the main trends already pointed out with the EE metric. Namely, with a green to NIR illumination change, Acriva Trinova’s optical performance shifts from a balanced energy distribution among its three foci to a predominance of its intermediate focus, AT Lisa Tri’s far focus benefit while intermediate and near foci are dimmed, and finally, Tecnis Symfony is, in practice, no longer a bifocal lens but a monofocal one.

## 4. Discussion

Despite the number of publications that deal with the problem of measuring the optical performance of diffractive IOLs under NIR light, there is still a considerable quantity of recent research work that overlook the results found in these publications. The risk of reaching the wrong conclusions becomes a reality when the visual performance of patients implanted with diffractive IOLs is analyzed by means of optical devices such as aberrometers and double-pass instruments which use NIR wavelengths for the said objective assessment [20,21,23,52].

In this work, we have demonstrated a clear mismatch between DMIOL performances measured with VIS (green) light and those measured with NIR illuminations. Such a mismatch depends strongly on the operative diffraction orders used in the IOL design and affect all the foci, that is, not only the near and intermediate, but also the far focus. We have simulated and experimentally measured the TF-EE with green (530 nm) and NIR (780 nm) wavelengths of three DMIOLs that use a different diffraction order (m) to conform their far focus: Acriva Trinova (m = −1), AT Lisa Tri (m = 0) and Tecnis Symfony (m = +1). Although aberrometers and double-pass objective clinical devices tend to focus on distance vision (i.e., the far focus of the DMIOL), our analysis provides a broader picture by including the lens performance from far to near. There are manifest changes in the optical performance from green to NIR that depend on the DMIOL design (Figure 3). First, there is a chromatic focal shift with NIR illumination which, if left uncorrected, would lead to erroneous refraction correction. Second, and more important, there is a strong variation in the energy that is directed to each of the foci of the IOLs. Thus, Acriva Trinova changed from a balanced energy distribution among its three foci, to an intermediate dominance at the expense of the far and the near foci. In the case of AT Lisa Tri, the far focus benefits from NIR illumination, while the near and, more significantly, the intermediate focus, experience a reduction in their EE. A more drastic variation in the optical performance between green and NIR illumination was found in the case of Tecnis Symfony. As such, with green light the IOL behaved as a bifocal low-addition lens while, under NIR illumination, it behaved as a monofocal lens.

To explain these results, let us assume a model for a hybrid refractive-diffractive IOL, according to which a thin carrier lens of refractive index *n_L_(λ)* with a diffractive lens profile engraved on one side is immersed in a medium of refractive index *n_A_(λ)*, which represents both the aqueous and vitreous humors. The optical power *P* of such hybrid refractive-diffractive IOL, at the specific design wavelength *λ_0_* (typically 550 nm), results from the addition of two terms, the refractive *P_r_* and the diffractive *P_d_* power (the latter usually referred to as the add power of the IOL),
*P*(λ_0_) = *P_r_*(*λ*_0_) + *P_d_*(*λ*_0_).(1)

We assume in Equation (1) that the lens thickness is negligible in comparison with the focal lengths involved.

The diffractive profile, designed for optimized performance at wavelength *λ_0_*, induces a phase shift (ϕ_0_) at every step edge that is proportional to its height (*h*), according to the expression
ϕ_0_ = (2π/λ_0_) · (*n_A_*(λ_0_) − *n_L_*(λ_0_)) *h*.(2)

Each focus of the IOL is formed with the contribution of one diffraction order, and thus, the diffractive lens shows m diffraction order powers, given by
*P_d_*(*λ_0_*, *m*) = *m · P_d_*(*λ*_0_,1).(3)

A given diffraction order contributes with *P_d_*(*λ_0_*, *m*) to the total power of the lens (Equation (1)) in a particular focus. We recall that, according to Equation (3), for a single diffractive lens (i.e., not yet combined with a refractive carrier lens), *negative* orders are divergent, *positive* orders are convergent, and the 0th diffraction order has null power. So, compared to green light, under NIR illumination the far focus of Acriva Trinova which is based on the m = −1 diffraction order, shifts towards the hyperopic side of the defocus curve, the far focus of AT Lisa Tri (m = 0 diffraction order) does not shift and the far focus of Tecnis Symfony (m = +1 diffraction order) shifts to the myopic side (Figure 3). Interestingly, it has been recently shown that autorefractors using NIR illumination provide statistically significant more myopic outcomes than objective refraction in patients implanted with the Tecnis Symfony IOL [4,53]. It would be very interesting to carry out the same clinical assessment in a cohort of patients implanted with Acriva Trinova to investigate if the autorefractor provides a more hyperopic refraction in this case. Therefore, to overcome the potential refraction error prediction bound to the NIR chromatic shift with a particular IOL design, one should introduce a priori in the measuring device (either autorefractor, aberrometer or double-pass instrument) which diffraction order is contributing to the far focus.

The step height of the diffractive lens produces an optical path difference (OPD) in lambda units of *p = (n_L_(λ_0_)-n_A_(λ_0_))h/λ_0_*. The OPD plays a central role in how the energy is split among the diffraction orders and thus, in the energy efficiency of the lens’ foci. For instance, in the case of kinoform (usually referred to as blazed sawtooth) diffraction profiles [54], on which Tecnis Symfony is based [25,55], for an illumination wavelength *λ* the diffraction energy efficiency of the m order is
η_m_ = sinc^2^ · (α*p* − m),(4)
where sinc x = sin(πx)/πx and α is defined as the fraction of 2π phase shift introduced for illumination wavelengths other than *λ_0_*. In the context of the eye, α is given by
α = (λ_0_/λ) · [(*n_A_*(λ) − *n_L_*(λ)) − (*n_A_*(λ_0_) − *n_L_*(λ_0_))].(5)

In Equation (5), the factor in brackets accounts for the influence of material dispersion. Since the IOLs in our study and the surrounding media have almost parallel dispersion curves [13,56], Equation (5) is frequently approximated by *α* ≈ *λ_0_/λ* and the OPD at a wavelength other than *λ_0_* becomes *p’= α p*. In the case of Tecnis Symfony, at the design wavelength *λ_0_* (550 nm), the step height *h* of the diffractive steps in the central zone of the IOL (3 diffractive rings) produces an OPD *p* = 1.5 to ideally deviate a similar amount of energy (40.5%) to the *m* = +1 and *m* = +2 diffraction orders (far and intermediate foci, respectively) for a balanced 50:50 energy distribution. On the other hand, the peripheral zone (6 diffractive rings) includes diffractive steps with OPD *p*= 1.366 to split more energy into the *m* = +1 diffraction order (63%) than in *m* = +2 diffraction order (21%) for a 75:25 energy distribution [25]. However, with NIR illumination (*λ* = 780 nm) the OPD *p*’ in the central and periphery zones of the lens reaches values very close to 1 (1.06 and 0.96, respectively). When *p*’ = 1, a widely known result is obtained: the lens reaches maximum efficiency, ideally 100%, in its *m* = +1 order (main focus) while other orders (and associated focus) vanish. That is why, with Tecnis Symfony, our experimental and simulated results of EE with 780 nm (Figure 3) show an optical performance that would closely approximate a monofocal IOL.

Analogous reasoning can be used to justify the fact that in the case of AT Lisa Tri and Acriva Trinova, the change from green to NIR illumination leads to an increase in the diffraction efficiency of the order m = 0, to the detriment of the rest of the diffraction orders. Therefore, both the far focus of the AT Lisa Tri and the intermediate focus of the Acriva Trinova, benefit from NIR illumination.

Although our experimental results with NIR illumination were obtained with the wavelength of 780 nm, which is typically used in double-pass instruments, other clinical equipment use even longer wavelengths deeper into the infrared. For instance, the Pyramidal aberrometer used by Alió et al. [21] employs an 850 nm central wavelength diode to illuminate the retina and capture the outcoming wavefront. The effect of using a longer wavelength should not give a significantly different result from those obtained with 780 nm because the dispersion curves of the optical materials commonly follow an almost hyperbolic shape, where the scaling of the refractive index with wavelength decreases as the wavelength increases [57]. Figure 7 shows the simulated TF-EE curves for 780 nm and 850 nm illuminations and the three DMIOLs with 4.5 mm IOL-pupil.

The minute differences between the simulated results with both the 780 nm and 850 nm NIR wavelengths imply that the conclusions of this study are valid for NIR illumination longer than 780 nm and, therefore, can be extended to a broader range of clinical studies [15,16,17,18,19] that used NIR-based instrumentation to objectively asses the visual quality of patients implanted with DMIOLs. For instance, in the case of DMIOLs that rely on either m = 0 or m = +1 diffraction orders for their far focus (such as AT Lisa Tri or Tecnis Symfony), an enhanced performance of this focus would be predicted while the near/intermediate focus would be underestimated [21,23]. Indeed, that was the case in an objective pseudo-accommodation analysis carried out with a double-pass system working at the NIR wavelength of 780 nm in patients implanted with diffractive bifocal IOLs, where the authors reported ‘monofocal-wise’ defocus curves (i.e., curves with only one visual acuity peak of far vision) [18]. With IOL designs such as Acriva Trinova that uses diffraction orders m = −1 and m = 0 and m = +1 for far, intermediate and near foci respectively, NIR-based assessment would likely lead to predicting detrimental far and near visual outcomes with enhanced intermediate performance.

To sum up, compensation for the refractive error bound to chromatic aberration is necessary with NIR-based instruments that provide objective refraction to avoid systematic refraction errors in patients implanted with DMIOLs. Otherwise, either myopic or hyperopic outcomes, respectively, are expected with IOLs that rely on *unconventional* diffraction orders m = +1 (Tecnis Symfony) and m = −1 (Acriva Trinova) for far vision. However, this compensation is not enough because it only considers the best focus shift but does not take into account the distribution of light energy among the foci, which strongly depends on the wavelength-dependent energy efficiency of the diffractive orders used in the DMIOL design. As such, with NIR illumination the optical imaging performance of the IOL’s foci becomes severely altered in comparison to VIS light. Therefore, the visual quality assessment of a diffractive pseudophakic eye based on NIR measurements should not be used to predict its performance under VIS illumination; otherwise, it may have inherent flaws that would lead to wrong conclusions depending on the particular IOL design. Clinicians should bear in mind that all these changes may lead to further deficient or contradictory interpretation of other assessments, such as defocus curves, contrast sensitivity and potential dysphotopsia (unwanted photic phenomena) experienced by the subjects after surgery.

Finally, it is worth noting that, to our knowledge, none of the clinical instruments currently used to measure the optical quality of patients implanted with DMIOLs handles any of the problems described above in their user manuals [58].

## Figures and Tables

**Figure 1 diagnostics-13-01259-f001:**
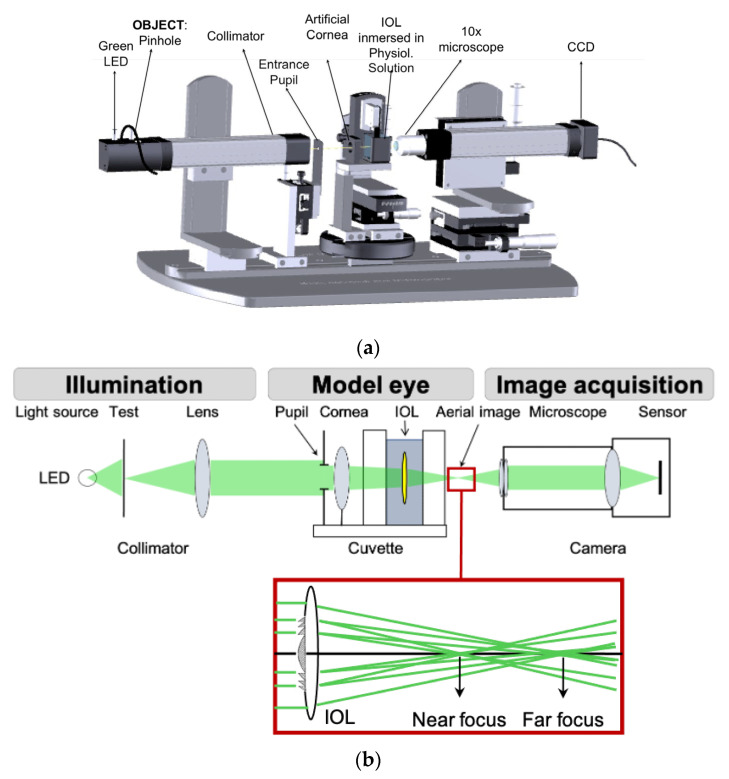
Optical setup used for in vitro assessment of the optical performance with VIS (green, 530 nm) and NIR (780 nm) illumination of the IOLs: Acriva Trinova, AT Lisa Tri and Tecnis Symfony. (**a**) General view, (**b**) layout of the optical setup. Inset: Far and near foci formed by a bifocal diffractive IOL. LED stands for light emitting diode.

**Figure 2 diagnostics-13-01259-f002:**
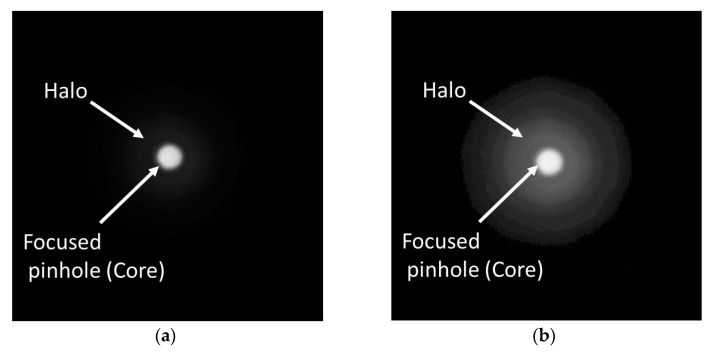
(**a**) Image of the pinhole test obtained with green light at the far focus of the Acriva Trinova IOL. A faint halo is barely discernible. The energy in the core and halo regions were obtained from these images (see text for details). (**b**) Same image as (**a**) but with gamma correction of 0.45 to enhance the visibility of the halo in the background of the image.

**Figure 3 diagnostics-13-01259-f003:**
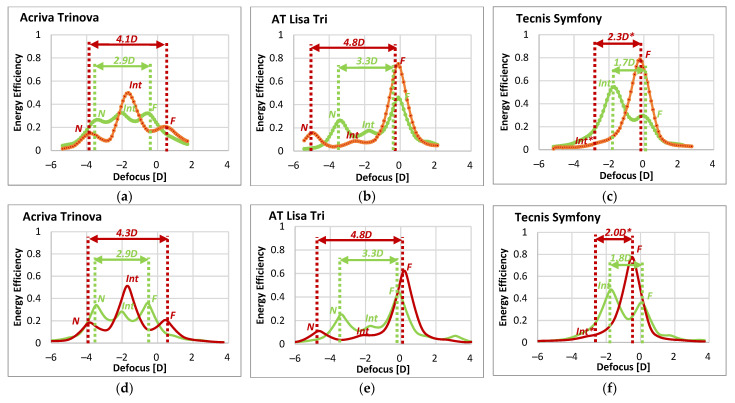
EE versus defocus (D) for the studied IOLs with 4.5 mm IOL-pupil. Upper row (**a**–**c**): Experimental results (**-●-●-** green, **-●-●-** NIR). Lower row (**d**–**f**): simulated results (green and red solid lines represent green and NIR illuminations, respectively). Far (*F*), intermediate (*Int*) and near (*N*) foci are labeled for each lens and illumination. The power difference (D) between the most distant foci (*F* to *N* for Acriva Trinova and AT Lisa Tri, *F* to *Int* for Tecnis Symfony) is marked with double arrows. Since Tecnis Symfony behaved monofocally under NIR, the position of the *Int ** focus as well as the experimental (2.3D *) and simulated (2.0D *) power differences were estimated (see text for details).

**Figure 4 diagnostics-13-01259-f004:**
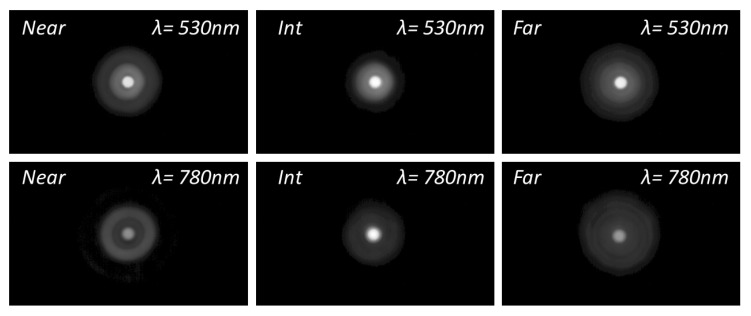
Experimental images of a pinhole test formed with Acriva Trinova IOL. Pupil 4.5 mm. Gamma Correction 0.45.

**Figure 5 diagnostics-13-01259-f005:**
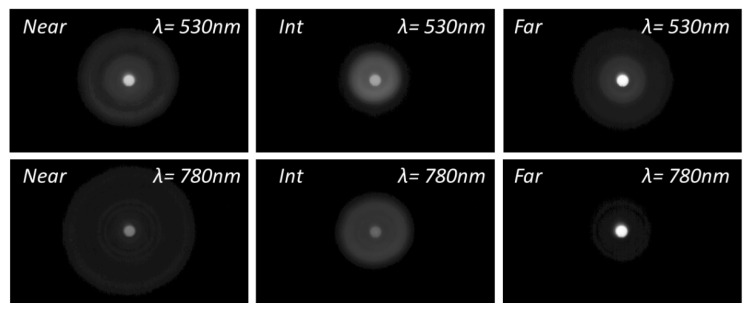
Experimental images of a pinhole test formed with AT Lisa Tri IOL. Pupil 4.5 mm. Gamma Correction 0.45.

**Figure 6 diagnostics-13-01259-f006:**
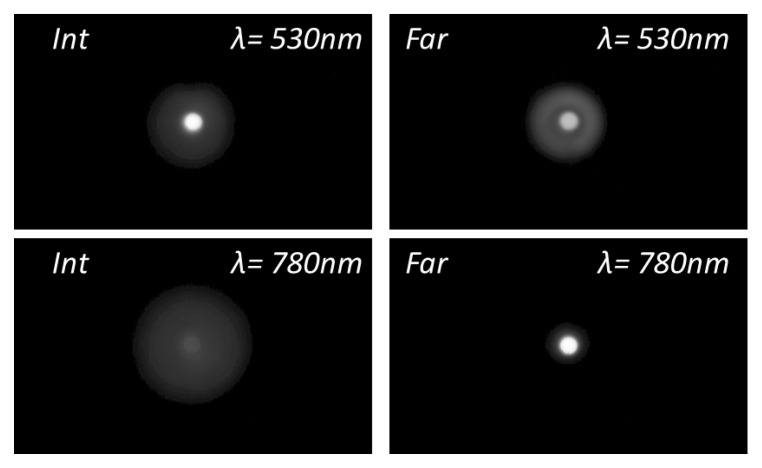
Experimental images of a pinhole test formed with Tecnis Symfony IOL. Pupil 4.5 mm. Gamma Correction 0.45.

**Figure 7 diagnostics-13-01259-f007:**
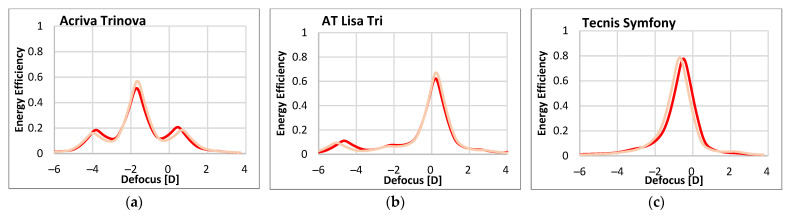
Simulated EE versus defocus for the studied IOLs with 4.5 mm IOL-pupil. (**a**) Acriva Trinova, (**b**) AT Lisa Tri and (**c**) Tecnis Symfony. Results were calculated for two NIR wavelengths: 780 nm (red line) and 850 nm (pink line).

**Table 1 diagnostics-13-01259-t001:** Specifications of the IOLs.

IOL Name	Refractive Index (550 nm)	Abbe Number	SA (µm)	Focus & Diffraction Order
Acriva Trinova	1.46	58	−0.165	Far: m = −1; Interm: m = 0; Near m = +1
AT Lisa Tri	1.46	56.5	−0.18	^(1)^ Far: m = 0; Interm: m = +1 ^(2)^ Far: m’ = 0; Near: m’ = +1
Tecnis Symfony	1.47	55	−0.27	Far: m = +1; Interm: m = +2

SA: spherical aberration (for a 6.0 mm entrance pupil). ^(1)^ First diffraction profile. ^(2)^ Second diffraction profile.

**Table 2 diagnostics-13-01259-t002:** Energy efficiency (EE) peak values obtained under green and NIR illuminations in the foci of the DMIOLs, either experimentally (Exper.) or by numerical simulation (Simul.).

	Acriva Trinova (*m* = −1) ^ⱡ^		AT Lisa Tri (*m* = 0) ^ⱡ^		Tecnis Symfony (*m* = +1) ^ⱡ^	
	EE_green_	EE_NIR_	EE_green_	EE_NIR_	EE_green_	EE_NIR_
	Exper./Simul.	Exper./Simul.	Exper./Simul.	Exper./Simul.	Exper./Simul.	Exper./Simul.
**Far focus**	0.32/0.36	0.21/0.21	0.46/0.44	0.75/0.62	0.30/0.36	0.79/0.78
**Intermediate focus**	0.33/0.28	0.50/0.51	0.17/0.15	0.09/0.08	0.55/0.48	0.07 */0.08 *
**Near focus**	0.27/0.34	0.15/0.18	0.26/0.25	0.16/0.11	NA	NA
**Power Difference (D)**	2.9 ^a^/2.9 ^a^	4.1 ^a^/4.3 ^a^	3.3 ^a^/3.3 ^a^	4.8 ^a^/4.8 ^a^	1.7 ^b,^*/1.8 ^b,^*	2.3 ^b,^*/2.0 ^b,^*

^ⱡ^ Diffraction order operating in the far focus of the lens. ^a^ Power difference between far and near foci. ^b^ Power difference between far and intermediate foci. * Estimated at −2.5D defocus. This defocus position corresponds to the theoretical add power of the intermediate focus of Tecnis Symfony under 780 nm NIR illumination. NA, not applicable.

## Data Availability

In addition to the Appendix A, authors declare their willing to provide with further data supporting the reported results upon reasonable request.

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
