# Peer review of "Pitfalls of Using NIR-Based Clinical Instruments to Test Eyes Implanted with Diffractive Intraocular Lenses"

_diagnostics, 2023, doi:10.3390/diagnostics13071259_

Round 1
Reviewer 1 Report
It was very disappointing to get to the end of the manuscript and finally find some semblance of a detailed explanation for what the current problem is with aberrometer and double pass NIR measurements used to assess correct positioning of diffractive multifocal intraocular lens (DFIOLs) during surgery to ensure maximal image contrast/quality at near, intermediate or far object distances. Lines 436-442 needed to come first in the introduction, as the authors failed multiple times to explicitly define important terminology in the main text, in addition to clearly stating what the problem is with these NIR devices that ophthalmologists use to aid in their IOL implant surgeries or evaluate surgical success post-surgery. Also, if these measurements are being acquired by clinicians post-surgery, please specify and facilitate understanding for your readers.
Naming a few examples, the author’s used phrasing like “induced a change” (what change? line 53-54), “overcome the aforementioned limitation of wavefront sampling” (what limitation?), “wavelength dependency” (What does this mean in the context of evaluating DFIOL performance?), and “to assess the optical quality” (how do you define optical quality? See line 71. Strehl ratio is only mentioned once, but this could be one definition). You also used “we emphasize that those analysis” without clearly stating what you mean by “analysis”. The readers are left to either infer or guess at what the authors mean or are referred to several other paper (which isn’t as helpful as they may think). Doing this you lose the opportunity to plainly say what your motivation for the study is, the problem you want to address, and how it will be done.
As the manuscript stands now, it’s unclear to me what the authors are trying to convey to clinicians with their findings. They present no evidence demonstrating how their measurements of the image PSF at different focal planes collected using their experimental setup at VIS and NIR wavelengths relates and affect the collected NIR light wavefronts of aberrometer and double-pass systems made by cataract surgeons. There isn’t even any speculation of how clinicians should deal with this issue of measuring wavefronts of diffractive IOLs using NIR light where there is an obvious chromatic focal shift that isn’t consistent with the wavelengths IOLs are designed for (i.e., visible wavelengths). Hypothetically, if cataract surgeons could take accurate wavefront measurements with visible light of the axially extended PSF from a DFIOL, what would a surgeon need to do differently in their positioning of the new IOL for that subject? How do your EE measurements of NIR light relate to how surgeons are currently interpreting their NIR aberrometer measurements, and what can be done about this? It’s clear that the authors are concerned about this issue of clinicians using NIR light for such methods, and rightly so. However, the experiments described, and results presented here do not aid in clarifying and/or providing instructions on how to rectify this procedural error that cataract surgeons are apparently committing and unaware of. I think a possible study the authors could consider is demonstrating how clinicians could compensate for the difference in chromatic focal shift between NIR and VIS light, while still using their NIR devices. I think this kind of information could be potentially useful and impactful on the field.
Additionally, only one object distance was analyzed in this study. Have you considered how the energy efficiency of the two to three diffractive order foci PSFs would change for VIS and NIR near, intermediate and far object distances? Only a single far object distance was presented (collimated beam imparted on the model eye), but how do intermediate and near object stimuli affect the diffraction efficiencies of these orders? Do your EE measurements for near foci increase with a near stimulus object? Knowing this may help determine first how well the diffractive order foci of these IOLs are performing for their intended object distances, and then, subsequently assess how “off” the NIR devices are in their measurements that cataract surgeons are using (at least for the three DFIOLs presented). This could then lead to a possible path that clinicians could take to correct this procedural error and improve surgical DFIOL implantation success.
Another note:
The authors present a contradiction in their definitions of light-in-the-bucket (LIB) metric and image PSF energy efficiency (EE) metric. They define LIB as the ratio of an aberrated wavefront and that of a monofocal diffraction limited PSF (see line 176), but later set LIB equal to EE, which are two seemingly different image evaluation criteria. The center core of your experimentally acquired image PSF is normalized to all the pixel intensities in your experimentally obtained image PSF. In the LIB metric, the amplitude of the PSF peak in the simulated LIB metrics should scale in amplitude for the monofocal diffraction limited PSF, while the simulated aberrated PSF contains sidelobes, and isn’t just considering the central core. It’s unclear how these relate, and I’m not convinced that LIB and EE can be set equal to each other.
Reviewer 2 Report
Dear Authors,
I read with interest the paper and I appreciated a lot the study design and the accuracy of the drafting.
According to my professional competencies, as a clinician, I have few questions about the clinical relevance of the paper:
- you performed the study using model eyes . What do you know about its clinical relevance ?
-how do you suggest to overcome these pitfalls of NIR based instruments?
Round 2
Reviewer 1 Report
From the author's comments, it makes it even more apparent that the author's logic is flawed in how they designed their simulations/experiments.
Your simulation and experimental protocol is ONLY useful for evaluating the image PSF of an object in the FAR FIELD because you are illuminating with collimated light or a planar wavefront. In fact, it's NOT SURPRISING that the NEAR PSF of the Bifocal DMIOL is distorted because this diffraction order of the lens that provides a near focus is designed to focus light coming from NEAR field/stimuli (e.g., divergent light), not collimated light or light coming from infinity that is planar. This is the flaw in the logic of your experimental/simulation design I want to bring to your attention. It is meaningless to evaluate the near focus PSF of these lenses without putting your source/object at near (which you did not do in this study). Thus, you need your experimental setup to reflect the real world scenario of light coming from both far distances and near distances to produce a meaningful study to evaluate focal plane PSFs.
Let me be clear, your optical system as you described, "The optical bench used complies with the International Organization for Standardization (ISO 11979-2) and allows for energy efficiency measurements at several pupils and focal planes. This type of set up is extensively employed in the field to obtain image quality metrics (MTF, cross correlation coefficients, energy efficiency…) and therefore to rank and assess the optical performance of different IOL designs" is only good for measuring PSFs (at whatever focal plane you want) for a object at FAR, and does a good job at that. You need to modify your setup to be able to measure PSFs for a near stimulus/object. Doing this will allow you to determine with confidence if these IOLs are refracting light well at near and far object distances.
